# 0.5-V 281-nW Versatile Mixed-Mode Filter Using Multiple-Input/Output Differential Difference Transconductance Amplifiers

**DOI:** 10.3390/s24010032

**Published:** 2023-12-20

**Authors:** Fabian Khateb, Montree Kumngern, Tomasz Kulej

**Affiliations:** 1Department of Microelectronics, Brno University of Technology, Technická 10, 601 90 Brno, Czech Republic; khateb@vutbr.cz; 2Faculty of Biomedical Engineering, Czech Technical University in Prague, nám. Sítná 3105, 272 01 Kladno, Czech Republic; 3Department of Electrical Engineering, Brno University of Defence, Kounicova 65, 662 10 Brno, Czech Republic; 4Department of Telecommunications Engineering, School of Engineering, King Mongkut’s Institute of Technology Ladkrabang, Bangkok 10520, Thailand; 5Department of Electrical Engineering, Czestochowa University of Technology, 42-201 Czestochowa, Poland; kulej@el.pcz.czest.pl

**Keywords:** universal filter, mixed-mode filter, differential difference transconductance amplifier, operational transconductance amplifier

## Abstract

This paper presents a new low-voltage versatile mixed-mode filter which uses a multiple-input/output differential difference transconductance amplifier (MIMO-DDTA). The multiple-input of the DDTA is realized using a multiple-input bulk-driven MOS transistor (MI-BD-MOST) technique to maintain a single differential pair, thereby achieving simple structure with minimal power consumption. In a single topology, the proposed filter can provide five standard filtering functions (low-pass, high-pass, band-pass, band-stop, and all-pass) in four modes: voltage (VM), current (CM), transadmittance (TAM), and transimpedance (TIM). This provides the full capability of a mixed-mode filter (i.e., twenty filter functions). Moreover, the VM filter offers high-input and low-output impedances and the CM filter offers high-output impedance; therefore, no buffer circuit is needed. The natural frequency of all filtering functions can be electronically controlled by a setting current. The voltage supply is 0.5 V and for a 4 nA setting current, the power consumption of the filter was 281 nW. The filter is suitable for low-frequency biomedical and sensor applications that require extremely low supply voltages and nano-watt power consumption. For the VM low-pass filter, the dynamic range was 58.23 dB @ 1% total harmonic distortion. The proposed filter was designed and simulated in the Cadence Virtuoso System Design Platform using the 0.18 µm TSMC CMOS technology.

## 1. Introduction

Active analog blocks, such as the operational amplifier (OA) or the transconductance amplifier (TA), are essential components for electronic devices, communication systems, and sensor interfaces. These blocks typically use the standard two inputs (i.e., a single differential stage). However, it has been confirmed that the use of a block with multiple inputs can reduce the number of components, silicon area, and power dissipation of some applications by a factor of approximately k, where k is the number of TA inputs [1]. Several applications based on this concept have been presented in [1,2,3,4]. Some other examples of multiple-input blocks are the differential difference amplifier (DDA) [5,6,7,8,9], differential difference current conveyor (DDCC) [10,11], differential difference operational floating amplifier (DDOFA) [12], differential difference transconductance amplifier (DDTA) [13,14], and many others. All these blocks allow for more arithmetic operations due to their multiple-input capabilities and are therefore widely used in instrumentation amplifiers, signal conditioning, differential amplification, filters, and many other applications. Although these blocks can reduce an application’s complexity and the number of blocks utilized, their internal structure is more complex than that of a standard two-input block. This is primarily due to the increased number of differential stages that are required to increase the number of inputs. The multiple-input MOS transistor (MI-MOST) provides a solution to avoid this problem and maintain a single differential stage [15,16,17]. It can be used in any standard CMOS technology without constraints. The first experimental results of MI-MOST are presented in [15,16,17] and various applications based on MI-MOST are presented in [18,19,20,21,22,23,24].

Filters play an important role in electronic, telecommunication and control systems. They can be used to reduce harmonics and filter noise in an electronic system, to separate or select desired signals, to remove unwanted signals in telecommunication systems, or to reduce the noise component of measurement signals in a control system. There are five common filtering functions that can be classified, namely the low-pass filter (LPF), high-pass filter (HPF), band-pass filter (BPF), band-stop filter (BSF), and all-pass filter (APF). These filtering functions can be designed using passive and active components, called passive filters or active filters, respectively. Second-order filters (or biquad filters) can be used to realize high-order filters applied to a high-fidelity three-ways crossover loudspeaker network and to a phase-locked loop.

Using active device-based filters, second-order LPF, HPF, BPF, BSF and APF (five filter functions) can be provided in a single topology, creating the so-called universal filter. Circuits that can provide voltage-mode (VM) (input and output as voltage), current-mode (CM) (input and output as current), transadmittance-mode (TAM) (input as voltage and output as current) and transimpedance-mode (TIM) (input as current and output as voltage) transfer functions in the same circuit are classified as mixed-mode universal filters. In a perfect mixed-mode universal filter, each mode of the transfer function should provide five filter functions, therefore obtaining twenty filter functions in a single topology. In addition, perfect universal filters should have high input impedance and low output impedance if the input and output are in voltage forms and low input and high output impedance if the input and output are in current forms.

There are many mixed-mode universal filters available in the literature [25,26,27,28,29,30,31,32,33,34,35,36,37,38,39,40,41,42,43,44,45,46,47,48,49,50,51,52,53,54,55,56,57,58,59,60,61]. The circuits in [25,26,27,28,29,30,31,32,33,34,35,36,37,38] realize a mixed-mode universal filter using variant active devices such as current conveyors [25,26,27,28,29,30,31,32,33,34], the CFOA (current feedback operational amplifier) [35,36,37], the FTFN (four terminal floating nullor) [38]; however, these filters lack electronic tuning capabilities. The circuits in [39,40,41] use current-controlled current conveyor-based filters to offer electronic tuning capability, but the circuits in [39,40,42] do not provide twenty transfer functions and the circuits in [41,42] require input matching conditions.

To obtain electronic tuning capability, the circuits in [43,44,45,46] use the CCTA (current conveyor transconductance amplifier), the circuit in [47] uses the VDTA (voltage differencing transconductance amplifier), the circuits in [48,49] use the VD-DVCC (voltage differencing differential voltage current conveyor), and the circuits in [50,51] use the VDBA (voltage differencing buffered amplifier). However, the circuits in [43,46,47] do not offer twenty transfer functions, the circuits in [48,49] require active/passive component matching conditions, and the circuits in [50,51] apply input voltage signals via a passive capacitor and/or resistor.

The OTA (operational transconductance amplifier) has been used to realize mixed-mode universal filters [52,53,54,55,56,57,58,59,60]. However, the circuits in [52,54,59] require passive or active components, the circuits in [53,56,57] do not provide twenty transfer functions, and the circuits in [52,53,58,60] require inverted input signals. It should be noted that the structure of active devices used in [25,26,27,28,29,30,31,32,33,34,35,36,37,38,39,40,41,42,43,44,45,46,47,48,49,50,51,52,53,54,55,56,57,58,59,60] is not designed for low-voltage low-power filters. Filters for such applications are in high demand, especially for biosignal and sensor signal processing. Many filters based on multiple-input DDTA have been presented [61,62,63,64,65,66,67,68].

This paper presents a versatile mixed-mode filter using MIMO-DDTAs. The circuit has six input voltages, three input currents, three output voltages, and two output currents; as such, it offers 61 transfer functions of LPF, BPF, HPF, BSF, and APF in the same topology. The six input voltage terminals possess a high-impedance level, and the three output voltage nodes possess a low-impedance level, which is ideal for voltage-mode circuits. The two output current terminals also possess a high-impedance level which can be connected directly to loads without buffer circuit requirements. The natural frequency of the filters can also be controlled electronically. The proposed versatile mixed-mode filter uses a supply voltage of 0.5 V and 281 nW of power consumption.

The paper is organized as follows: Section 2 describes the multiple-input/output DDTA. Section 3 describes the application of the versatile mixed-mode filter and non-ideality analysis. Section 4 presents the simulation results. Finally, the conclusion is given in Section 5.

## 2. Proposed DDTA Circuit with Multiple-Input and Multiple-Output

The electrical symbol of the proposed multiple-input/output differential-difference transconductance amplifier is shown in Figure 1. Its performance, in an ideal case, is described by Equation (1). The circuit possesses one low-impedance output w, which provides a difference of the sums of the voltages *V_y_*_+_ and *V_y_*_−_, applied to its non-inverting and inverting terminals, respectively. It further has a high-impedance output o, which provides a current, proportional to the voltage *V_w_* appearing at the w terminal.
(1)$$\left.\begin{array}{c} V_{w}=V_{y+1}+V_{y+2}-V_{y-1}-V_{y-2}\\ I_{o\pm }=\pm g_{m}V_{w} \end{array}\right\}$$

The CMOS structure of the proposed circuit is shown in Figure 2. The circuit consists of two blocks, a multiple-input differential-difference amplifier (MI-DDA) and a multiple-output transconductance amplifier (MO-TA).

The MI-DDA can be seen as a two-stage internal OTA, operating in a unity-gain feedback configuration. The first gain stage is formed by the transistors M_1_–M_12_, M_14_, M_15_, while the second stage is formed by the transistors M_13_ and M_16_. The capacitance C_C_ is used for frequency compensation. The first stage can be seen as a current-mirror OTA, with a differential amplifier M_1_–M_10_ and a set of current mirrors M_5_–M_12_, M_6_–M_11_, M_14_–M_15_, acting as a differential to a single output converter.

The input stage is based on a non-tailed bulk-driven differential pair M_1_-M_4_, which behaves as a differential amplifier with high CMRR and PSRR performances, while also being able to operate at extremely low supply voltages [69], even lower than the threshold voltages of the used MOS transistors. In order to increase the voltage gain, a partial positive feedback (PPF) is applied. The PPF is created by two cross-coupled transistor pairs: M_7_–M_8_ and M_9_–M_10_. The cross-coupled pairs generate negative conductances that partially compensate for the conductances of the diode-connected transistors M_2A,B_ for the “upper” pair, and M_5_, M_6_ for the “lower” pair. Therefore, the resulting conductances increase at the drains of these transistors, and, consequently, the first stage transconductance and voltage gain also increase. In particular, the upper pair increases the voltage gain from the bulk terminals to the gates of M_1A,B_ [70], while the lower pair increases the current gains of the current mirrors M_5_–M_12_ and M_6_–M_11_ [71]. The combination of two PPF circuits decreases the overall sensitivity of the transconductance gain of the first stage to transistor mismatch [63]. This achieves a larger voltage gain while maintaining relatively low sensitivity of the input stage and avoiding problems with frequency compensation of the DDA.

In order to realize a differential to difference function without duplicating the input stage, the multiple inputs were realized using the so-called multiple-input BD MOS transistors [15]. The symbol and the implementation of the devices are shown in Figure 3a,b, respectively. A passive capacitive voltage divider is applied to the bulk terminal of the MOS transistor, thus creating a multiple-input device. The large resistors R_MOSi_, used to bias the bulk terminal for DC, are realized using two minimum-size MOS transistors operating in a cut-off region, as shown in Figure 3c.

Assuming 1/ω*C_Bi_* << R_MOSi_, the voltage V_b_ at the bulk terminal of the MI-BD-MOS transistor can be expressed as:(2)$$V_{b}=\sum_{i=1}^{n}\beta_{i}V_{i}$$
where n is the number of inputs and *β_i_* is the voltage gain of the input capacitive divider:(3)$$\beta_{i}=\frac{C_{Bi}}{\sum_{i=1}^{n}C_{Bi}}$$

Note that with equal *C_Bi_*, *β_i_* = 1/n.

The open-loop voltage gain of the DDA can be expressed as:(4)$$A_{vo}=\beta \frac{2g_{mb1}\left(r_{ds15}||r_{ds12}\right)g_{m16}\left(r_{ds16}||r_{ds13}\right)}{\left(1-m_{1}\right)\left(1-m_{2}\right)}$$
where the coefficients *m*_1_ and *m*_2_ are the ratios of the absolute values of the negative and positive conductances in a lower and upper PPF circuit, respectively [62]:(5)$$m_{1}=\frac{g_{m9,10}}{g_{m5,6}+g_{ds2}+g_{ds3,4}+g_{ds7,8}}≅\frac{g_{m9,10}}{g_{m5,6}}$$
(6)$$m_{2}=\frac{g_{m7,8}}{g_{m2}+g_{ds1}+g_{ds5,6}+g_{ds9,10}}≅\frac{g_{m7,8}}{g_{m2}}$$

Note that the above coefficients should always be lower than unity to maintain circuit stability. In the proposed design, *m*_1_ = *m*_2_ = 0.5. This increased the voltage gain by 12 dB, thus compensating for the gain loss introduced by the input capacitive divider (approximately 10 dB) while maintaining the overall circuit sensitivity to transistor mismatch at a relatively low level.

The second block creating the MIMO-DDTA is the multiple output transconductance amplifier. The circuit can be seen as a current-mirror linear OTA. Note that a version of the MI-TA with one positive output was presented and verified experimentally in [18]. Here, a second, inverting output has been added, thus increasing the circuit universality. Transistors M_1_, M_2_ and M_11_, M_12_ realize an input differential stage. The transistors M_11_ and M_12_ operate in a triode region and extend the linear range of the structure. The circuit can be seen as a BD version of the Krummenacher and Joehl transconductor [72], operating in weak inversion. Thanks to the BD approach, the linear range of the circuit is extended η = *g_m_*_1,2_/*g_mb_*_1,2_ times, as compared with its gate-driven (GD) counterpart. In order to obtain optimum linearity, the following condition should be met [18]:(7)$$k=\frac{{\left(W/L\right)}_{11,12}}{{\left(W/L\right)}_{1,2}}=0.5$$
where *W* and *L* are the MOS transistor channel width and length, respectively.

Assuming unity current gain of all current mirrors, the circuit transconductance is given by:(8)$$g_{m}=\eta \frac{4k}{4k+1}\cdot \frac{I_{set}}{n_{p}U_{T}}$$
where *n_p_* is the subthreshold slope factor, *U_T_* is the thermal potential and *I_set_* is the biasing current. Note that the circuit transconductance is proportional to this current.

In order to increase the DC voltage gain of the structure while not limiting its output voltage range, all current mirrors are based on self-cascode transistors. Consequently, the DC voltage gain from the input to the differential output is equal to:(9)$$A_{VO}≅{2g}_{m}[\left(g_{m9}r_{ds9}r_{ds9c}\right)||\left(g_{m6}r_{ds6}r_{ds6c}\right)]$$

Thanks to the self-cascode technique, it is possible to compensate for the gain loss associated with the application of the BD technique. In practice, a voltage gain of around 40 dB can be obtained.

## 3. Versatile Mixed-Mode Filter

Figure 4 shows the proposed versatile mixed-mode universal filter employing four MIMO-DDTAs and two grounded capacitors. Using (1) and nodal analysis, the output voltages *V_o_*_1_, *V_o_*_2_, *V_o_*_3_ and the output currents *I_o_*_1_, *I_o_*_2_ can be given by:(10)$$V_{o1}=\frac{g_{m3}}{g_{m4}}\cdot \frac{s^{2}C_{1}C_{2}\left(V_{5}-V_{6}\right)+{sC}_{1}g_{m2}\left(V_{3}-V_{4}\right)+g_{m1}g_{m2}\left(V_{1}-V_{2}\right)}{s^{2}C_{1}C_{2}+{sC}_{1}g_{m3}+g_{m2}g_{m3}}$$
(11)$$V_{o2}=\frac{s^{2}C_{1}C_{2}\left(V_{5}-V_{6}\right)+{sC}_{1}g_{m2}\left(V_{3}-V_{4}\right)+g_{m1}g_{m2}\left(V_{1}-V_{2}\right)}{s^{2}C_{1}C_{2}+{sC}_{1}g_{m3}+g_{m2}g_{m3}}$$
(12)$$V_{o3}=\frac{{sC}_{2}g_{m3}\left(V_{5}-V_{6}\right)+\left(s^{2}C_{1}C_{2}+{sC}_{2}g_{m3}\right)\left(V_{3}-V_{4}\right)+\left({sC}_{2}g_{m3}+g_{m1}g_{m3}\right)\left(V_{1}-V_{2}\right)}{s^{2}C_{1}C_{2}+{sC}_{1}g_{m3}+g_{m2}g_{m3}}$$
(13)$$I_{o1}=g_{m3}\cdot \frac{s^{2}C_{1}C_{2}\left(V_{5}-V_{6}\right)+{sC}_{1}g_{m2}\left(V_{3}-V_{4}\right)+g_{m1}g_{m2}\left(V_{1}-V_{2}\right)}{s^{2}C_{1}C_{2}+{sC}_{1}g_{m3}+g_{m2}g_{m3}}$$
(14)$$I_{o1}=\frac{-s^{2}C_{1}C_{2}I_{3}{+sC}_{1}g_{m3}I_{2}-g_{m2}g_{m3}I_{1}}{s^{2}C_{1}C_{2}+{sC}_{1}g_{m3}+g_{m2}g_{m3}}$$
(15)$$I_{o2}=\frac{s^{2}C_{1}C_{2}I_{3}{-sC}_{1}g_{m3}I_{2}+g_{m2}g_{m3}I_{1}}{s^{2}C_{1}C_{2}+{sC}_{1}g_{m3}+g_{m2}g_{m3}}$$
(16)$$V_{o1}=\frac{1}{g_{m4}}\cdot \frac{-s^{2}C_{1}C_{2}I_{3}{+sC}_{1}g_{m3}I_{2}-g_{m2}g_{m3}I_{1}}{s^{2}C_{1}C_{2}+{sC}_{1}g_{m3}+g_{m2}g_{m3}}$$

From (10)–(16), the variant filtering functions can be determined and are shown in Table 1. The proposed mixed-mode universal filter can offer LP, HP, BP, BS, and AP filtering functions of VM, CM, TAM, and TIM in the same topology. Thanks to the multiple inputs of the DDTA, the VM, CM, and TIM can offer non-inverting and inverting transfer functions of LP, HP, BP, BS, and AP filters, and the VM and TAM can also offer differential transfer functions of LP, HP, BP, BS, and AP filters. Thus, the proposed mixed-mode filter can provide 61 transfer functions in a single topology. Thanks to the multiple outputs of the DDTA, such as DDTA_4_, the proposed filter utilizes a minimum number of used DDTAs while offering inverting and non-inverting transfer functions of LP, HP, BP, BS, and AP filters of CM. The input signals *V*_1_ to *V*_6_ are connected to the high-impedance terminals of the DDTA; thus, the voltage signals can be applied without any buffer circuit requirements. The output signals *V_o_*_1_ to *V_o_*_3_ are connected to the low-impedance terminals of the DDTA, which offers a cascadable output for the voltage-mode filter structures. The output signals *I_o_*_1_ and *I_o_*_2_ are connected to the high-impedance terminals of the DDTA, which offers a cascadable output for the current-mode filter structures. However, in the case of CM, the inputs *I*_1_ to *I*_3_ require additional circuits, such as multiple-output current followers or multiple-output current mirrors, to create three identical current signals from the single original current signal. It is clear that the proposed filter is exempt from inverting-type input signal and input matching conditions for realizing all filtering functions both in the case of voltage and current signals.

The voltage gain *g_m_*_3_/*g_m_*_4_ of the filtering functions can be obtained if the output *V_o_*_1_ is used. In the case of TAM, the inputs *V*_1_ to *V*_6_ are converted to output currents by *g_m_*_3_; in the case of TIM, the input currents *I*_1_ to *I*_3_ are converted to output voltages by *g_m_*_4_.

The natural frequency (*ω_o_*) and the quality factor (*Q*) can be given by:(17)$$\omega_{o}=\sqrt{\frac{g_{m2}g_{m3}}{C_{1}C_{2}}}$$
(18)$$Q=\sqrt{\frac{C_{2}g_{m2}}{C_{1}g_{m3}}}$$

It should be noted that the parameter *ω_o_* can be controlled electronically by *g_m_*_2_ and *g_m_*_3_ and the parameter *Q* can be given by *C*_2_/*C*_1_.

### Non-Ideality Analysis

Taking the tracking errors and the non-ideal transconductance of the MIMO-DDTA into account, the characteristics of the MIMO-DDTA can be rewritten as:(19)$$\left.\begin{array}{c} V_{w}={\alpha_{j+}V}_{y+1}+\alpha_{j+}V_{y+2}-\alpha_{j-}V_{y-}+\alpha_{j-}V_{y-2}\\ I_{o}=g_{mnj}V_{w} \end{array}\right\}$$
where *α_j+_* =1–*ε_j+v_* and *ε_j+v_* (|*ε_j+v_* |≪ 1) denote the voltage tracking error from non-inverting terminals (i.e., *V_y_*_+1_, *V_y_*_+2_) to the w-terminal (i.e., *V_w_*) of the *j*-th DDTA, *α_j_*_−_ =1–*ε_j-v_* and *ε_j-v_* (|*ε_j-v_*|≪ 1) denote the voltage tracking error from inverting terminals (i.e., *V_y_*_−1_, *V_y_*_−2_) to the w-terminal (i.e., *V_w_*) of the *j*-th DDTA, and *g_mnj_* is the non-ideal transconductance gain of the *j*-th DDTA. The non-ideal transconductance *g_mnj_* of the *j*-th DDTA at a frequency near the cut-off frequency can be expressed by [59]:(20)$$g_{mnj}\left(s\right)≅g_{mj}\left(1-\mu_{j}s\right)$$
where *μ_j_*= 1⁄*ω_gmj_*, *ω_gmj_* denotes the first pole frequency of the *j*-th *g_m_*.

Using (19), the denominator of (10)–(16) can be modified as:(21)$$s^{2}C_{1}C_{2}+sC_{1}g_{mn3}\alpha_{3-}+g_{mn2}g_{mn3}\alpha_{2+}\alpha_{3-}$$

Using (20), (21) becomes:(22)$$\left\{s^{2}C_{1}C_{2}\left(1-\frac{C_{1}g_{m3}\mu_{3}\alpha_{3-}-{g_{m2}g_{m3}\alpha_{2+}\alpha_{3-}\mu }_{2}\mu_{3}}{C_{1}C_{2}}\right)+sC_{1}g_{m3}\alpha_{3-}\left(1-\frac{g_{m2}g_{m3}\alpha_{2+}\alpha_{3-}\left(\mu_{2}+\mu_{3}\right)}{C_{1}g_{m3}\alpha_{3-}}\right)+g_{m2}g_{m3}\alpha_{2+}\alpha_{3-}\right\}$$

The tracking errors and the non-ideal effect of the transconductance of the DDTA can be made negligible by satisfying the following condition:(23)$$\left.\begin{array}{c} \frac{g_{m2}g_{m3}\alpha_{2+}\alpha_{3-}\left(\mu_{2}+\mu_{3}\right)}{C_{1}g_{m3}\alpha_{3-}}\ll 1\\ \frac{C_{1}g_{m3}\mu_{3}\alpha_{3-}-{g_{m2}g_{m3}\alpha_{2+}\alpha_{3-}\mu }_{2}\mu_{3}}{C_{1}C_{2}}\ll 1 \end{array}\right\}$$

The modified natural frequency (*ω_on_*) and the modified quality factor (*Q_n_*) can be expressed as:(24)$$\omega_{on}=\sqrt{\frac{g_{m2}g_{m3}}{C_{1}C_{2}}\cdot \alpha_{2+}\alpha_{3-}}$$
(25)$$Q_{n}=\sqrt{\frac{C_{2}g_{m2}}{C_{1}g_{m3}}\cdot \frac{\alpha_{2+}}{\alpha_{3-}}}$$

To consider the parasitic impedances that affect the proposed mixed-mode filter, the parasitic capacitance *C_o_* and parasitic conductance *g_o_* (*g_o_* = 1/*R_o_*, *R_o_* is the output resistance) at the o-terminal of the DDTA are considered while the parasitic impedances at the y- and w-terminals are neglected. Considering Figure 4, the parasitic capacitances *C_o_*_1_, *C_o_*_4_ and parasitic conductances *g_o_*_1_, *g_o_*_4_ are parallel with *C*_1_ and the parasitic capacitances C_o2_, C_o4_, and parasitic conductances *g_o_*_2_, *g_o_*_4_ are parallel with *C*_2_. *C_o_*_1_, *C_o_*_2_, *C_o_*_4_ are, respectively, the parasitic capacitances at the o-terminal of DDTA_1_, DDTA_2_, DDTA_4_, and *g_o_*_1_, *g_o_*_2_, *g_o_*_4_ are, respectively, the parasitic conductances at the o-terminal of DDTA_1_, DDTA_2_, DDTA_4_. The parasitic capacitances can be neglected by appropriately choosing values such that *C*_1_ ≫ *C_o_*_4_ + *C_o_*_4_, *C*_2_ ≫ *C_o_*_2_ + *C_o_*_4_, *g_m_*_2_ ≫ *g_o_*_1_ + *g_o_*_4_, and *g_m_*_3_ ≫ *g_o_*_2_ + *g_o_*_4_.

## 4. Simulation Results

The circuit was designed and simulated using the Cadence Virtuoso System Design Platform using 0.18 µm CMOS technology from TSMC. The voltage supply was ±250 mV (0.5 V) and the bias voltage V_B1_ = −100 mV. The transistor aspect ratios are included in Table 2. It is worth noting that the only increase in chip area is due to the input capacitor C_B_ = 0.5 pF, so the total input capacitance of the proposed MIMO-DDTA is 3 pF. This value is acceptable for integration.

Selected simulation results for the MIMO-DDTA are shown in Figure 5 and Figure 6. Figure 5 shows the simulated results of the DC transfer characteristic of the MI-DDA *V_w_* versus *V_y+_*_1_ and of the MO-TA *I_o_*_+_, *I_o_*_−_ versus *V_w_* with various *I_set_*. The extended linearity of operation despite the low supply voltage is observed.

Figure 6 shows the impedances frequency characteristics of the MIMO-DDTA with *I_set_* = 4 nA: (**a**) Z_y_, Z_o+,_ Z_o−_ and (**b**) Z_W_. At low frequency, the impedance of Z_y_ = 29.5 GΩ, Z_o+_ =Z_o−_ = 2.1 GΩ and Z_w_ = 876 Ω. All these values are suitable for the proposed filter application.

For the filter application, for cutoff frequency 220 Hz and for *I_set_* = 4 nA (*g_m_* = 27.7 nS) the Equation (17) has been used to calculate the value of capacitors C_1_ = C_2_ = 20 pF. The frequency responses of the gain and phase for the differential input VM, non-inverting CM, TAM and TIM filter with *I_set_*_1–4_ = 4 nA are shown in Figure 7. The simulated cutoff frequency was 211 Hz, which is closed to the calculated one. This slight deviation in the cutoff frequency can be easily corrected by adjusting the setting current. The power consumption of the filter was 281 nW.

The frequency responses of LP, HP, BP, BS, and AP gains and phases for VM are shown in Figure 8. The wide tunability of the filter is achieved by varying the setting current *I_set_*_1–4_ = (0.5, 1, 2, 4) nA, where the cutoff frequency was (28, 56, 112, 211) Hz, respectively.

Monte Carlo (MC) analysis was used to perform the statistical analysis to estimate the parametric yield and generate information about the performance characteristics of the differential input VM filter. The gains frequency responses of LP, HP, BP, BS, and AP with 200 runs MC are shown in Figure 9. The curves are overlapping or close to each other.

The process, voltage, and temperature (PVT) corners were also used to confirm the robustness of the design. The process transistor corners were fast–fast, fast–slow, slow–fast, and slow–slow. The process MIM capacitor corners were fast–fast and slow–slow. The voltage supply corners were = ±10% (V_DD_-V_SS_) and the temperature corners were −20 °C and 70 °C. The results for the gains frequency responses of LP, HP, BP, BS, and AP with PVT are shown in Figure 10. The curves are again overlapping or close to each other, which confirms the robustness of the filter design. In addition, thanks to the tunability of the filter, any deviation in the cutoff frequency can be easily adjusted by the setting current.

The transient response of the VM LPF with an applied input sinusoidal signal V_in-pp_ = 200 mV@10 Hz is shown in Figure 11a. The spectrum of the output signal is shown in Figure 11b, where the total harmonic distortion (THD) of 0.23% is indicated.

The THD for the VM LPF with different peak-to-peak input signal values @ 10 Hz is shown in Figure 12. The 1% THD is achieved for V_in-pp_ = 300 mV. The output voltage noise for the VM LPF is shown in Figure 13. The root-mean-square (RMS) output noise integrated in the bandwidth of 1 to 211 Hz was 130 μV; thus, the dynamic range (DR) of the VM LPF filter is 58.23 dB @ 1% THD.

The proposed versatile mixed-mode filter was compared with the previously reported filters in [29,32,45,51,59,60,61] as shown in Table 3. Compared with these previous works, the proposed filter offers the most transfer functions of the five standard filtering functions and the lowest voltage supply. Compared with [29,32], the proposed filter offers electronic tuning capability of the natural frequency; compared with [59,60,61], the proposed filter uses fewer active devices. The filters in [32,45,51] apply the input signal via capacitor and/or resistor, the structure in [45] does not provide five standard filtering functions of VM, CM, TAM, and TIM, and the filters in [45,51] require input matching conditions for realizing some filtering functions.

## 5. Conclusions

This paper presents a 0.5 V, 281 nW versatile mixed-mode universal filter using MIMO-DDTAs. The MIMO-DDTA is used to realize a versatile mixed-mode universal filter that offers many transfer functions in the same topology. To realize variant transfer functions such as LPF, HPF, BPF, BSF, and APF of VM, CM, TAM, and TIM, inverted input signal requirement is absent. The natural frequency can be electronically controlled. The VM filter offers high-input impedance and low-output impedance, and the CM filter offers high-output impedance. For the VM LP filter, the dynamic range was 58.23 dB @ 1% total harmonic distortion. The proposed filter was designed and simulated in the Cadence Virtuoso System Design Platform using the 0.18 µm CMOS technology from TSMC. The simulation results, including Monte-Carlo and PVT corners, confirm the functionality of the design.

## Figures and Tables

**Figure 1 sensors-24-00032-f001:**
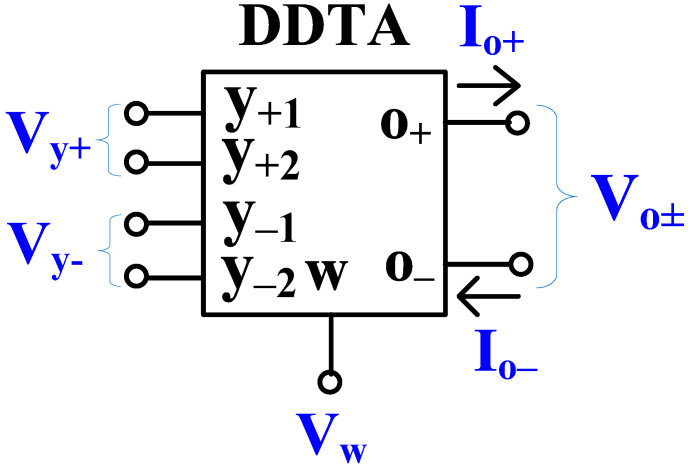
Electrical symbol of the MIMO-DDTA.

**Figure 2 sensors-24-00032-f002:**
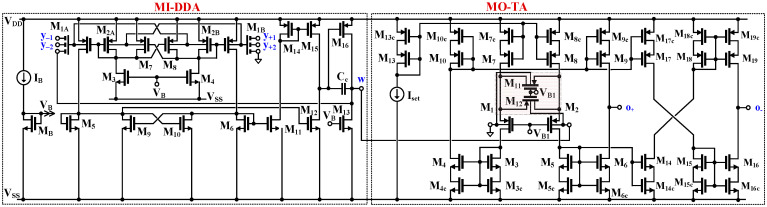
CMOS structure of the MIMO-DDTA.

**Figure 3 sensors-24-00032-f003:**
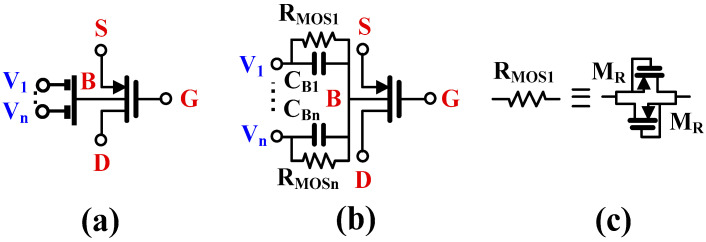
MI-BD MOST: (**a**) symbol, (**b**) possible implementation, (**c**) implementation of R_MOS_.

**Figure 4 sensors-24-00032-f004:**
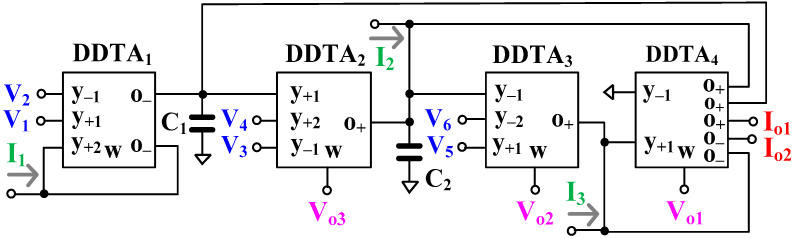
The proposed versatile mixed-mode filter using MIMO-DDTAs.

**Figure 5 sensors-24-00032-f005:**
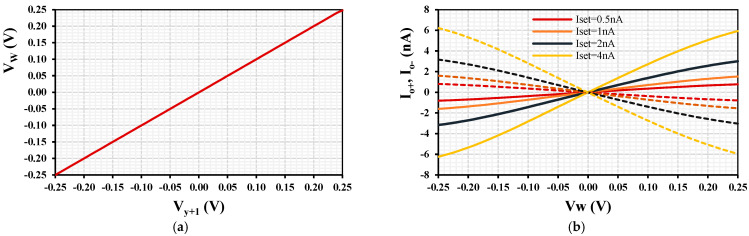
The DC transfer characteristics of the MIMO-DDTA: (**a**) *V_w_* versus *V_y+_*_1_ and (**b**) *I_o_*_+_, *I_o_*_−_ (dashed line) versus *V_w_* with various *I_set_*.

**Figure 6 sensors-24-00032-f006:**
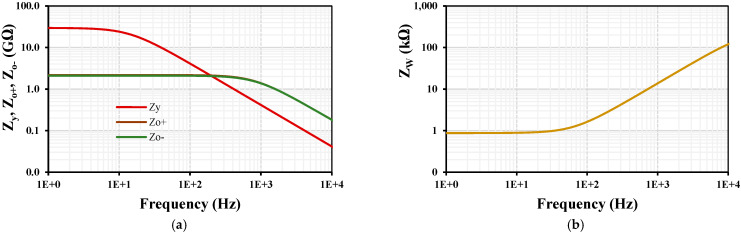
The impedances frequency characteristics of the MIMO-DDTA: (**a**) Z_y_, Z_o+,_ Z_o−_ and (**b**) Z_W_.

**Figure 7 sensors-24-00032-f007:**
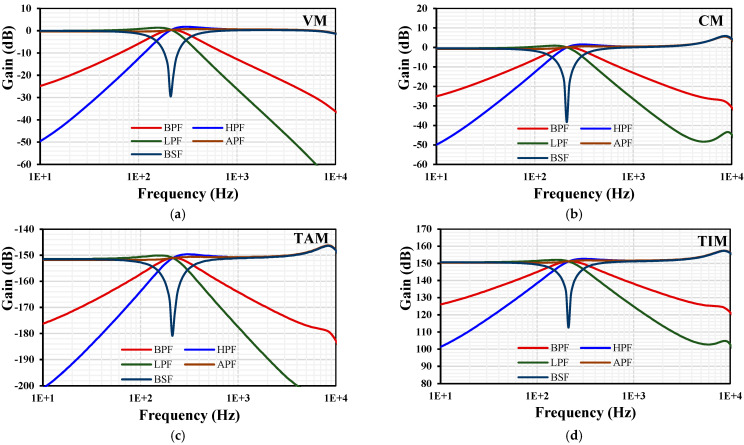
The frequency characteristics of gains for the VM (**a**), CM (**b**), TAM (**c**), and TIM (**d**).

**Figure 8 sensors-24-00032-f008:**
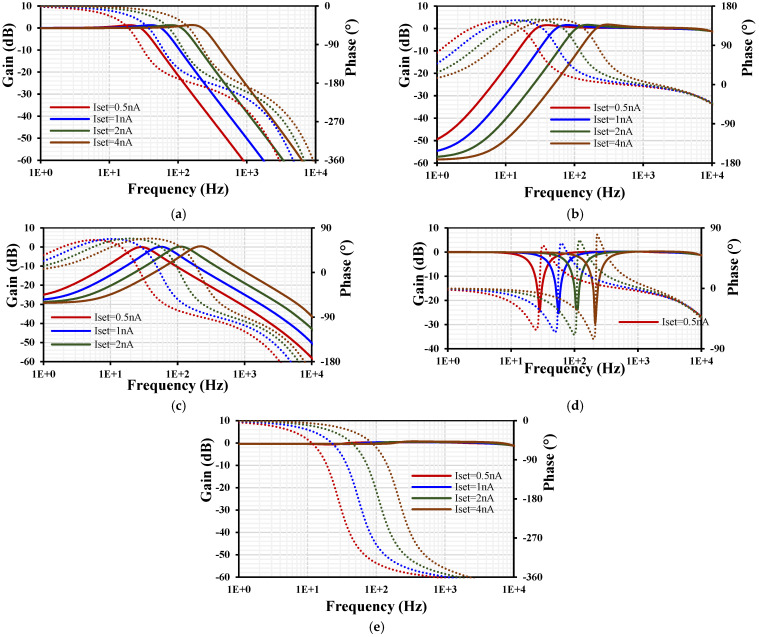
The frequency characteristics of gains (lines) and phases (points) for the VM filter: LPF (**a**), HPF (**b**), BPF (**c**), BSF (**d**), and APF (**e**).

**Figure 9 sensors-24-00032-f009:**
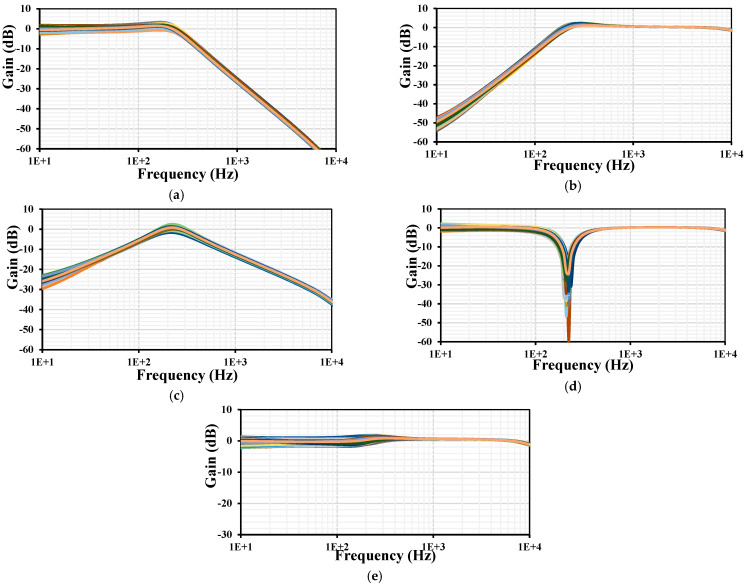
The 200 runs MC frequency characteristics of the gains for the differential input VM filter: LPF (**a**), HPF (**b**), BPF (**c**), BSF (**d**), and APF (**e**).

**Figure 10 sensors-24-00032-f010:**
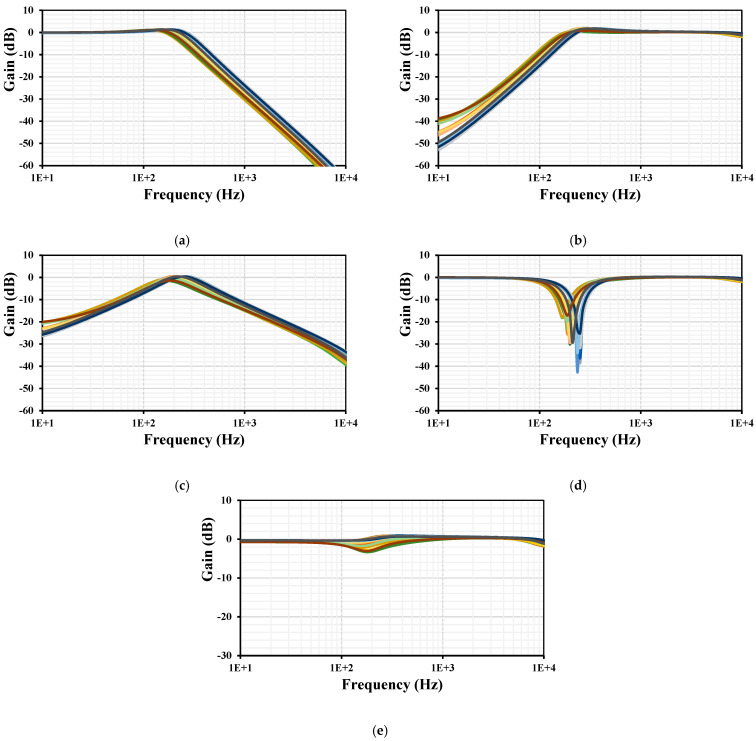
The PVT frequency characteristics of the gains for the VM filter: LPF (**a**), HPF (**b**), BPF (**c**), BSF (**d**), and APF (**e**).

**Figure 11 sensors-24-00032-f011:**
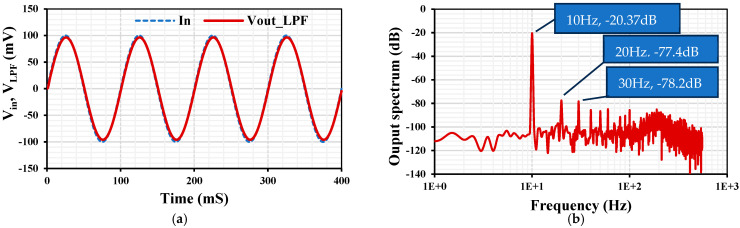
The transient response of the VM LPF (**a**) and the spectrum of the output signal (**b**).

**Figure 12 sensors-24-00032-f012:**
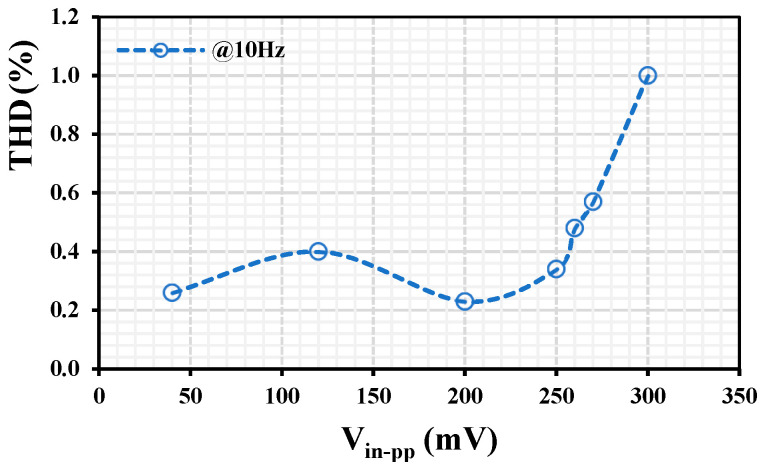
The THD of the VM LPF with different peak-to-peak input voltages @ 10 Hz.

**Figure 13 sensors-24-00032-f013:**
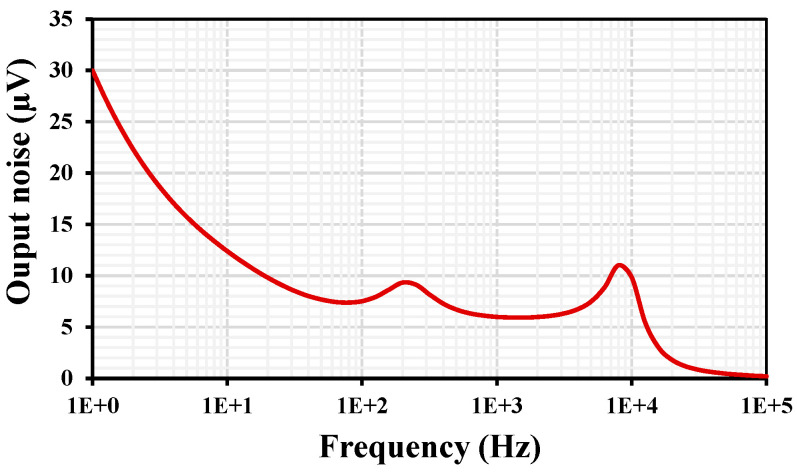
The output voltage noise of the VM LPF.

**Table 1 sensors-24-00032-t001:** Obtaining variant filtering functions of the proposed versatile mixed-mode filter.

Operation Mode	Filtering Function		Input	Output
VM	LP	Non-inverting	$V_{1}$	$V_{o1}$
		Inverting	$V_{2}$	$V_{o1}$
		Non-inverting	$V_{1}$	$V_{o2}$
		Inverting	$V_{2}$	$V_{o2}$
		Non-inverting	$V_{1}{=V}_{6}$	$V_{o3}$
		Inverting	$V_{2}{=V}_{5}$	$V_{o3}$
		Differential	$V_{1}{-V}_{2}$	$V_{o2}$
	BP	Non-inverting	$V_{3}$	$V_{o1}$
		Inverting	$V_{4}$	$V_{o1}$
		Non-inverting	$V_{3}$	$V_{o2}$
		Inverting	$V_{4}$	$V_{o2}$
		Non-inverting	$V_{5}$	$V_{o3}$
		Inverting	$V_{6}$	$V_{o3}$
		Differential	$V_{3}{-V}_{4}$	$V_{o2}$
	HP	Non-inverting	$V_{5}$	$V_{o1}$
		Inverting	$V_{6}$	$V_{o1}$
		Non-inverting	$V_{5}$	$V_{o2}$
		Inverting	$V_{6}$	$V_{o2}$
		Non-inverting	$V_{3}{=V}_{6}$	$V_{o3}$
		Inverting	$V_{4}{=V}_{5}$	$V_{o3}$
		Differential	$V_{5}{-V}_{6}$	$V_{o2}$
	BS	Non-inverting	$V_{1}{=V}_{5}$	$V_{o1}$
		Inverting	$V_{2}{=V}_{6}$	$V_{o1}$
		Non-inverting	$V_{1}{=V}_{5}$	$V_{o2}$
		Inverting	$V_{2}{=V}_{6}$	$V_{o2}$
		Differential	$\left(V_{1}{=V}_{5}\right)-\left(V_{2}{=V}_{6}\right)$	$V_{o2}$
	AP	Non-inverting	$V_{1}{=V}_{4}=V_{5}$	$V_{o1}$
		Inverting	$V_{2}{=V}_{3}=V_{6}$	$V_{o1}$
		Non-inverting	$V_{1}{=V}_{4}=V_{5}$	$V_{o2}$
		Inverting	$V_{2}{=V}_{3}=V_{6}$	$V_{o2}$
		Differential	$\left(V_{1}{=V}_{4}=V_{5}\right)-\left(V_{2}{=V}_{3}=V_{6}\right)$	$V_{o2}$
CM	LP	Non-inverting	$I_{1}$	$I_{o1}$
		Inverting	$I_{1}$	$I_{o2}$
	BP	Non-inverting	$I_{2}$	$I_{o2}$
		Inverting	$I_{2}$	$I_{o1}$
	HP	Non-inverting	$I_{3}$	$I_{o1}$
		Inverting	$I_{3}$	$I_{o2}$
	BS	Non-inverting	$I_{1}=I_{3}$	$I_{o1}$
		Inverting	$I_{1}=I_{3}$	$I_{o2}$
	AP	Non-inverting	$I_{1}=I_{2}=I_{3}$	$I_{o1}$
		Inverting	$I_{1}=I_{2}=I_{3}$	$I_{o2}$
TAM	LP	Non-inverting	$V_{1}$	$I_{o1}$
		Inverting	$V_{2}$	$I_{o1}$
		Differential	$V_{1}{-V}_{2}$	$I_{o1}$
	BP	Non-inverting	$V_{3}$	$I_{o1}$
		Inverting	$V_{4}$	$I_{o1}$
		Differential	$V_{3}{-V}_{4}$	$I_{o1}$
	HP	Non-inverting	$V_{5}$	$I_{o1}$
		Inverting	$V_{6}$	$I_{o1}$
		Differential	$V_{5}{-V}_{6}$	$I_{o1}$
	BS	Non-inverting	$V_{1}{=V}_{5}$	$I_{o1}$
		Inverting	$V_{2}{=V}_{6}$	$I_{o1}$
		Differential	$\left(V_{1}{=V}_{5})-(V_{2}{=V}_{6}\right)$	$I_{o1}$
	AP	Non-inverting	$V_{1}{=V}_{4}=V_{5}$	$I_{o1}$
		Inverting	$V_{2}{=V}_{3}=V_{6}$	$I_{o1}$
		Differential	$\left(V_{1}{=V}_{4}=V_{5}\right)-\left(V_{2}{=V}_{3}=V_{6}\right)$	$I_{o1}$
TIM	LP	Non-inverting	$I_{1}$	$V_{o1}$
	BP	Inverting	$I_{2}$	$V_{o1}$
	HP	Non-inverting	$I_{3}$	$V_{o1}$
	BS	Non-inverting	$I_{1}=I_{3}$	$V_{o1}$
	AP	Non-inverting	$I_{1}=I_{2}=I_{3}$	$V_{o1}$

**Table 2 sensors-24-00032-t002:** Transistor aspect ratios of the MIMO-DDTA.

**MI-DDA**	*W*/*L* (µm/µm)
M_1A_, M_2A_, M_1B_, M_2B_ M_14_, M_15_	16/3
M_3_–M_8_, M_11_–M_12_, M_B_	8/3
M_9_, M_10_	4/3
M_16_	6 × 16/3
M_13_	6 × 8/3
M_R_	4/5
MIM capacitor: C_B_ = 0.5 pF, C_c_ = 6 pF	
**MO-TA**	*W*/*L* (µm/µm)
M_1_, M_2_	2 × 15/1
M_3_–M_6_, M_14_–M_16_	2 × 10/1
M_3c_–M_6c_, M_14c_–M_16c_	10/1
M_7_–M_10_, M_17_–M_19_, M_13_	2 × 15/1
M_7c_–M_10c_, M_17c_–M_19c_, M_13c_, M_11_, M_12_	15/1

**Table 3 sensors-24-00032-t003:** Comparison of the proposed filter’s properties with those of mixed-mode universal filters.

Factor	Proposed	[29]	[32]	[45]	[51]	[59]	[60]	[61]
Number of active devices	4-DDTA	3-DDCC	1-FDCCII, 1-DDCC	2-VDBA	3-VDBA	5-OTA	8-OTA	5-DDTA
Realization	0.18 µm CMOS	0.25 µm CMOS	0.18 µm CMOS	0.18 µm CMOS	0.18 µm CMOS	0.18 µm CMOS	0.18 µm CMOS	0.18 µm CMOS
Number of passive devices	2-C	2-C, 3-R	2-C, 6-R	2-C, 2-R	2-C, 1-R	2-C	2-C	2-C
Type of filter	MIMO	MISO	MIMO	MIMO	MIMO	MISO	MIMO	MIMO
Total number of offered responses	61	30	36	17	20	20	20	36
Each mode offers five standard responses	Yes	Yes	Yes	No	Yes	Yes	Yes	Yes
Orthogonal control of $\omega_{o}$ and Q	Yes	Yes	Yes	Yes	Yes	Yes	Yes	Yes
Electronic control of $\omega_{o}$	Yes	No	No	Yes	Yes	Yes	Yes	Yes
All passive devices grounded	Yes	Yes	No	No	No	Yes	Yes	Yes
High input impedances for VM	Yes	Yes	No	No	No	Yes	Yes	Yes
No need for input matching conditions	Yes	Yes	Yes	Yes	No	Yes	Yes	Yes
No need for inverting input conditions	Yes	Yes	Yes	Yes	Yes	Yes	Yes	Yes
Power supply (V)	0.5	±1.25	±0.9	±0.75	±1.25	±0.9	±0.3	1.2
Power dissipation (mW)	0.281 × 10^−3^	-	-	0.373	5.482	0.1773	0.00577	0.33
Natural frequency (kHz)	0.211	3.315 × 10^3^	1.591 × 10^3^	1.44 × 10^3^	16.32 × 10^3^	3.39 × 10^3^	5	1.04
Total harmonic distortion (%)	1@300 mV_pp_ (LPF)	0.723@60 µA_pp_	2.2@300 mV_pp_	2.2@200 mV_pp_	<4@350 mV_pp_ (HPF)	-	2@120 mV_pp_ (LPF)	1.09@650 mV_pp_
Dynamic range (dB)	58.23	-	-	-	-	-	53.2	-
Verification of result	Sim	Sim	Sim	Sim/Exp	Sim/Exp	Sim	Sim	Sim/Exp

Note: MIMO = multiple-input multiple-output, MISO = multiple-input single-output.

## Data Availability

Data are contained within the article.

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
