# Peer review of "0.5-V 281-nW Versatile Mixed-Mode Filter Using Multiple-Input/Output Differential Difference Transconductance Amplifiers"

_sensors, 2023, doi:10.3390/s24010032_

Round 1

Reviewer 1 Report

Comments and Suggestions for Authors

This paper describes about the mixed mode filter using MIMO DDTA that the authors proposed, and the quality for the paper is quite good. however, the followings need to be clarified before publication.

1) Since the authors proposed the DDTA, this proposed DDTA needs to be evaluated at first compared with previously published articles in terms of performances such as gain, power consumption, stability, noise and etc.

2) For filters, they used C1 and C2 using 20pF which are little bit big based on the circuit structure, current consumption and bias voltage used. How these capacitor values were derived logically needs to be explained before realizing filters in more detail.

3) Did the authors perform layout for MIMO filter proposed? The readers may need the total area for the chip layout to decide if this MIMO filter can fit to some area constraints.

4) Is it quite reasonable to compare the circuit level simulation with some other articles showing measured results along with simulated ones?

5) They depicted the natural frequency and Q-factor . Can they present how the natural frequency and Q-factor were improved using the proposed method?

Comments on the Quality of English Language

Their quality of English is very good.

Author Response

Reviewer # 1

Comments and Suggestions for Authors

This paper describes about the mixed mode filter using MIMO DDTA that the authors proposed, and the quality for the paper is quite good. however, the followings need to be clarified before publication.

1) Since the authors proposed the DDTA, this proposed DDTA needs to be evaluated at first compared with previously published articles in terms of performances such as gain, power consumption, stability, noise and etc.

Author response:

Thank you for your positive review. Unfortunately, based on our research, we were unable to find any papers presenting the performance of a standalone DDTA, but the performance of an application based on it. This comparison is already done on Table 3.

However, we provided new simulation results of the proposed DDTA as shown in the new added figures 5 and 6. The simulation results confirms the extended linearity of operation despite the low supply voltage is observed.

Selected simulation results for the MIMO-DDTA are shown in Figs. 5 and 6. Fig. 5 shows the simulated results of the DC transfer characteristic of the MI-DDA (Vw versus Vy+1) and of the MO-TA (Io+, Io- versus Vw). The extended linearity of operation despite the low supply voltage is observed.

(a)

(b)

Figure 5. The DC transfer characteristics of the MIMO-DDTA: (a) Vw versus Vy+1 and (b) Io+, Io- (dashed line) versus Vw.

Figure 6. shows the impedances frequency characteristics of the MIMO-DDTA: a) Zy, Zo+ , Zo- and b) ZW. At low frequency the impedance of Zy=29.5GΩ, Zo+=Zo-=2.1GΩ and Zw=876Ω.

(a)

(b)

Figure 6. The impedances frequency characteristics of the MIMO-DDTA: (a) Zy, Zo+ , Zo- and (b) ZW.

2) For filters, they used C1 and C2 using 20pF which are little bit big based on the circuit structure, current consumption and bias voltage used. How these capacitor values were derived logically needs to be explained before realizing filters in more detail.

Author response:

The value of the capacitors was calculated based on equation 17 and for a given value of Iset=4nA (gm=27.7nS). The calculated value of the cutoff frequency was 220 Hz and the simulated value was 211 Hz. The two values are close to each other. The main advantage is the tunability of the filter, hence any unacceptable variation in cutoff frequency can be corrected by adjusting the setting current.

The information has been added to the manuscript.

In general, very low frequencies filters usually require large capacitors. If the problem of capacitance would be critical, active capacitance multipliers could be used or gm could be lowered at the cost of a larger input noise. For example, for C1=C2=5pF and Iset=2nA the frequency characteristic is shown below. However, the cutoff frequency is 398Hz. Therefore, to obtain lower cutoff frequency a 20pF were used. 

3) Did the authors perform layout for MIMO filter proposed? The readers may need the total area for the chip layout to decide if this MIMO filter can fit to some area constraints.

Author response:

Since the circuit was not planned for fabrication, no layout was made. However, the only increase in chip area is due to the input capacitor CB=0.5pF, so the total input capacitance of the proposed MIMO-DDTA is 3pF. This value is acceptable for integration.

The above information has been added to the manuscript. 

4) Is it quite reasonable to compare the circuit level simulation with some other articles showing measured results along with simulated ones?

Author response:

Thank you for your comment. Please note that only results from circuit-level simulations of some other blocks were used for the comparison in Table 3. The measured results are not used for comparison, but only to show the verification of the results.

5) They depicted the natural frequency and Q-factor . Can they present how the natural frequency and Q-factor were improved using the proposed method?

Author response:

Thank you for your comment. Based on Eqs. 17 and 18, the parameter ωo can be controlled electronically by gm2 and gm3 (i.e. Iset2 and Iset3) and the parameter Q can be given by C2/C1. The natural frequency that varied by gm (via setting current) are shown in Fig. 8.

However, it should be noted here that the main advantage of using MIMO-DDTA is that in a single topology, the proposed filter can provide five standard filtering functions (LP, HP, BP, BS and AP) in four modes: VM, CM, TAM and TIM. This ensures the full capability of the mixed-mode filter (i.e., twenty filter functions).

The text has been highlighted in the manuscript. 

Reviewer 2 Report

Comments and Suggestions for Authors

The paper is well written and the theory is sound

Author Response

Reviewer # 2

Comments and Suggestions for Authors

The paper is well written and the theory is sound

Author response:

Thank you for your positive evaluation of our work.

Reviewer 3 Report

Comments and Suggestions for Authors

- No conclusions are described. That section only shows some extra data.

- The BR topology presents high variations in its performance. The cause and possible recommendations may be discussed.

- The Layout is not presented. So, post-layout simulations were not included.

- The Nonideality analysis assumes the use of very large capacitors (20 pF). Then area consumption mus be calculated.

Author Response

Reviewer # 3

Comments and Suggestions for Authors

- No conclusions are described. That section only shows some extra data.

Author response:

Thank you for your comment. This section has been improved. Some text has been added as follows:

“To realize variant transfer functions such as LPF, HPF, BPF, BSF, and APF of VM, CM, TAM, and TIM, inverted input signal requirement is absent. The natural frequency can be electronically controlled. VM filter offers high-input impedance and low-output impedance, and CM filter offers high-output impedance.”

- The BR topology presents high variations in its performance. The cause and possible recommendations may be discussed.

Author response:

Thank you for your comment. The variation is expected because the circuit operates in subthreshold voltage, which is sensitive mainly to temperature, among other corners. However, as mentioned, the main advantage is the tunability of the filter, so any unacceptable variations in cutoff frequency can be corrected by adjusting the setting current Iset.

This information has been highlighted in the manuscript. 

- The Layout is not presented. So, post-layout simulations were not included.

Author response:

Since the circuit was not planned for fabrication, no layout was made.

It should be mentioned here that post layout simulation is important for circuits operating at high frequencies to determine the effect of layout parasitic capacitances on circuit performance. However, here we have a circuit operating at low and very low frequencies (<10 kHz), so the effect of parasitic capacitances could be negligible in this case. For such a circuit, the MC analysis and PVT corners are sufficient to demonstrate the functionality and the performance of the circuit.

- The Nonideality analysis assumes the use of very large capacitors (20 pF). Then area consumption mus be calculated.

Author response:

Thank you for your comments. In general, very low frequencies filters usually require large capacitors. If the problem of capacitance would be critical, active capacitance multipliers could be used or gm could be lowered at the cost of a larger input noise. For example, for C1=C2=5pF and Iset=2nA the frequency characteristic is shown below. However, the cutoff frequency is 398Hz. Therefore, to obtain lower cutoff frequency a 20pF were used. 

Round 2

Reviewer 1 Report

Comments and Suggestions for Authors

All issues are resolved, therefore it can be considered for publication in this form.